# Exploring *Dolichos lablab* compounds as potential inhibitors for fusion (F) protein of human metapneumovirus (HMPV): A systematic computational approach

Md. Mainuddin Hossain[1], Md. Jahid Hasan Apu[2], Md. Faisal Bin Abdul Aziz[3], Md. Tanzimur Rahman Tanjil[4], Liton Chandra Das[1], Antora Kar[1], Fatematuz Zuhura Evamoni[5], Md. Mahbub Morshed[6]*

1 Department of Biotechnology and Genetic Engineering, Mawlana Bhashani Science and Technology University, Tangail, Bangladesh, 2 Ministry of Public Administration, Dhaka, Bangladesh, 3 Department of Computer Science and Engineering, Comilla University, Comilla, Bangladesh, 4 Department of Chemistry, Mawlana Bhashani Science and Technology University, Tangail, Bangladesh, 5 Department of Biotechnology and Genetic Engineering, Noakhali Science and Technology University, Noakhali, Bangladesh, 6 Department of Pharmacy, Noakhali Science and Technology University, Noakhali, Bangladesh

* mahbub@student.nstu.edu.bd

## Abstract

One of the most crucial respiratory pathogens in the world, namely human metapneumovirus (HMPV), causes acute upper and lower respiratory tract infection. The HMPV Fusion (F) protein is a vital element for viral entry and is the sole target of neutralizing antibodies, making it a prime target for drug and vaccine development. Targeting the Fusion (F) protein of HMPV for inhibition has emerged as a potential therapeutic strategy, particularly in respiratory infection treatment. We aimed to identify potential inhibitors against HMPV F protein by molecular docking and molecular dynamics study. Through molecular docking, we were able to identify 16 lead compounds derived from *Dolichos lablab (DL)*. These compounds exhibited robust binding affinities with the HMPV F protein, with better docking scores compared to the ribavirin inhibitor as a control with a −6.7 kcal/mol docking score. Among these top-ranked compounds, Brassinolide (CID_115196), Quercetin (CID_5280343), and 2'-Hydroxygenistein (CID_5282074) demonstrated favorable molecular, pharmacokinetics, and drug-like properties, promising biological activities, and acceptable toxicity profiles. Furthermore, Brassinolide, Quercetin, and 2'-Hydroxygenistein were found to be promising drug inhibitors with the greatest binding stability against the HMPV F protein compared to the ribavirin inhibitor, which is validated by the highest protein-ligand interactions and lowest Root Mean Square Deviation (RMSD), Root Mean Square Fluctuation (RMSF), and Radius of Gyration (Rg) values using 100 ns molecular dynamic simulation. Our study provides valuable insights into the therapeutic potential of *DL* compounds as potential or hypothetical inhibitors for

**Data availability statement:** All relevant data are within the paper and its Supporting information files.

**Funding:** The author(s) received no specific funding for this work.

**Competing interests:** The authors have declared that no competing interests exist.

HMPV F protein having three promising candidates- Brassinolide, Quercetin, and 2'-Hydroxygenistein. These results warrant further validation through detailed in vitro and in vivo investigations.

## Introduction

Acute respiratory infections (ARIs) are among the leading causes of morbidity and mortality worldwide, accounting for a significant burden of disease, particularly among children under five years, the elderly, and immunocompromised individuals [1]. These infections are responsible for millions of hospitalizations annually and pose a continuous threat to global public health, especially in low- and middle-income countries [2]. While bacteria can be responsible for some ARIs, the majority are caused by respiratory viruses, including human metapneumovirus (HMPV), influenza virus, respiratory syncytial virus (RSV), and severe acute respiratory syndrome coronavirus 2 (SARS-CoV-2) [3–6]. These viruses are well-known for causing seasonal epidemics and, in some cases, pandemics, with high transmission rates and substantial healthcare impacts. In recent years, increasing attention has been directed toward the development of antiviral therapeutics targeting HMPV and related respiratory paramyxoviruses, such as RSV and influenza virus. Among these emerging pathogens is human metapneumovirus (HMPV), a negative-sense RNA virus from the Pneumoviridae family, which has been increasingly identified as a major cause of lower respiratory tract infections (LRTIs). HMPV infections are most prevalent in young children, especially those under five years of age, as well as elderly and immunocompromised individuals [7]. A study by Peiris et al. revealed that 5.5% of hospitalized children under 18 with respiratory tract infections tested positive for HMPV, with a mean age of 32 months [8]. A critical factor in HMPV pathogenesis is the fusion (F) protein, which mediates viral entry by binding to heparan sulfate (HS) and RGD-binding integrins on the host cell surface [9–11]. The HMPV F protein exists in pre-fusion and post-fusion conformations [12,13]. This study focused on the pre-fusion form, a metastable state crucial for viral entry and the primary target of neutralizing antibodies and antivirals. Our analysis aligns with current insights into the F protein's role in HMPV pathogenesis. Given its essential role in viral attachment and membrane fusion, the F protein has emerged as a promising target for therapeutic intervention and vaccine development. Natural products derived from plants, particularly phytochemicals, have been the primary source of potent drug candidates [14–17]. Phytochemicals have been employed for therapeutic purposes throughout history in the form of conventional medications, potions, and oils. World Health Organization (WHO) estimates that 122 plant-derived medications have implications for ethnopharmacology, and 80% of the world's population still uses traditional plant-derived medicines for basic healthcare. For instance, the well-known anti-inflammatory drug "aspirin" is produced from a natural substance. Additionally, digitoxin, an active plant-derived component, promotes the heart's ability to contract. Penicillin is also the most well-known natural substance made from a fungus [18]. Doxorubicin is used to treat both Hodgkins and non-Hodgkins lymphomas, as well as acute leukemia, lung and thyroid cancers, soft tissue and bone sarcomas

[17]. These plant-based phytochemicals are far less dangerous and safer than synthetic chemical compounds [19]. The preliminary pharmacological studies revealed that *Dolichos lablab* possessed antidiabetic, anti-inflammatory, analgesic, antioxidant, cytotoxic, hypolipidemic, antimicrobial, insecticidal, hepatoprotective, antilithiatic, antispasmodic effects and also used for the treatment of iron deficiency anemia [20].

Due to the absence of efficient antiviral compounds and their poor performance, environmentally friendly phytopharmaceuticals based on phytochemicals that prevent viral entry and replication while having affordable and tolerable side effects are required to treat viral infections [21,22] as well as there were no studies for understanding the role of bioactive compounds in DL to inhibit Fusion (F) proteins and regulate respiratory infection conditions. Therefore, we aimed to find efficient inhibitors and therapeutic targets from *Dolichos Lablab* (DL) for preventing the attachment and function of the fusion protein of HMPV. We have listed phytochemicals of DL through literature reviews and docked them against the fusion protein using a molecular docking technique that quickly determines the binding affinities and modes between the target substrate (such as protein) and a variety of ligands, such as phytochemicals. Pharmacokinetics, drug-like properties, and toxicity profile analysis were done by admetSAR, SwissADME, pKCSM, Deep-PK tools. Bioactivities of the drug candidates was predicted by Molinspiration tools, and lastly, molecular dynamics simulation was performed by Schrodinger. A detailed overview of the methodology is presented in Fig 1.

## Materials and methods

### Retrieval of *Dolichos Lablab*-derived phytochemicals (Ligands)

We retrieved *Dolichos Lablab* (DL)-derived phytochemicals from a database, namely PubChem. The National Institution of Health (NIH) administers the PubChem database, which mostly comprises small molecules but also includes larger compounds such as carbohydrates, lipids, peptides, nucleotides, and chemically engineered macromolecules [23]. 86 different DL-derived phytochemicals were retrieved from the PubChem database (S1 Table).

### Ligand preparation

We used SWISS PDB Viewer 4.1 software for energy minimization of our phytochemicals. SWISS PDB Viewer 4.1 software is a visualization software that includes energy minimization capabilities, and it can perform energy minimization tasks for small molecules and ligands developed by the Swiss Institute of Bioinformatics (SIB) [24].

### Retrieval of target protein and preparation

For the purpose of the target protein, we explored various literature reviews. To retrieve the crystal structure of our target protein, we explored a database named Research Collaboratory for Structural Bioinformatics Protein Data Bank (RCSB PDB). Education and research in basic biology, health, energy, and biotechnology depend on the global Protein Data Bank (PDB) database of 3D structure data for larger biological molecules such as proteins, DNA, and RNA, which has been stored at RCSB PDB in the United States [25].

We retrieved the crystal structure of the Fusion (F) protein of HMPV (PDB-ID: 7sej; resolution 2.51 Å) from RCSB PDB. The retrieved protein structure was capacitated and depleted through computation using the most recent versions of Discovery Studio 4.5. We removed all of the inhibitors, water molecules, and heteroatoms from HMPV through Discovery Studio 4.5. We also used SWISS PDB Viewer 4.1 software for energy minimization of the Fusion (F) protein of HMPV.

### Molecular docking studies

To evaluate the binding affinities between the *DL*-derived phytochemicals and HMPV fusion protein, we employed the PyRx virtual screening tool and Autodock Vina, v.1.2.0 [26] for molecular docking. It makes the binding pose clear by displaying every possible orientation and conformation for any specific ligand at the fusion protein and phytochemical binding site. The substrate-binding pocket that corresponds to the primary protease's active site was identified using a grid box in

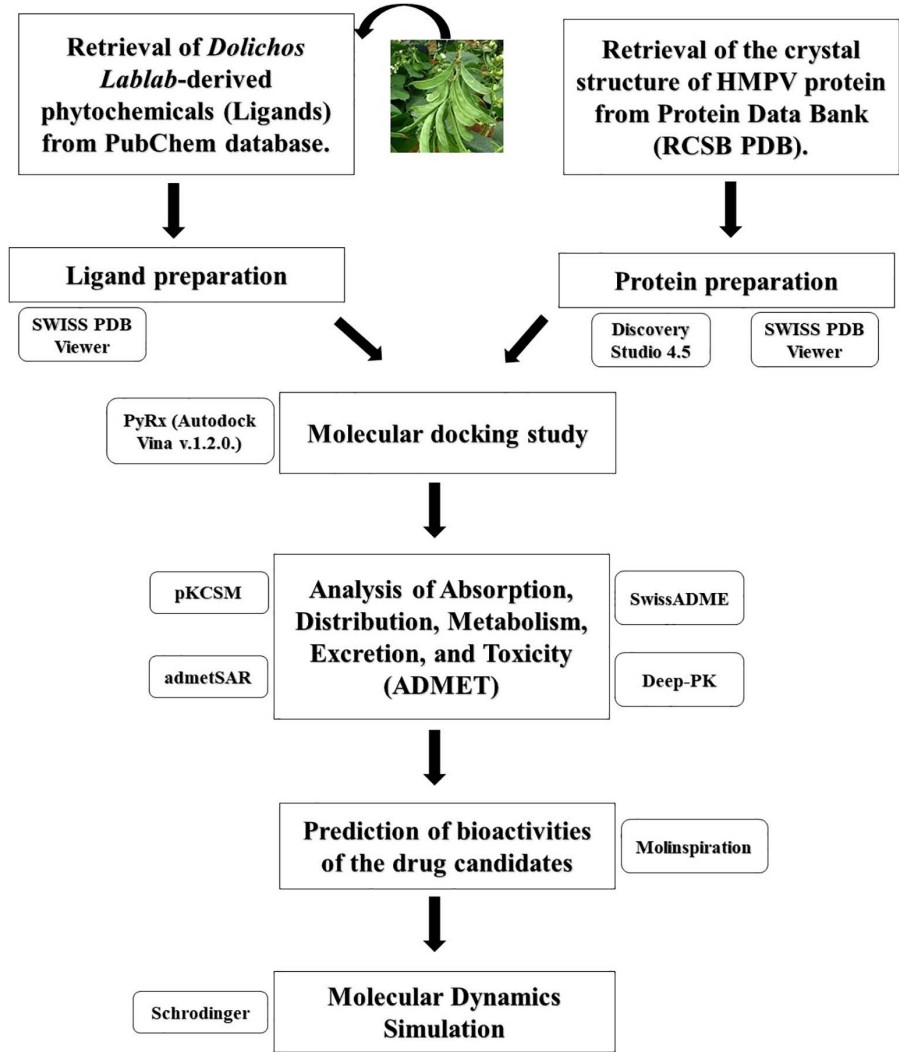

**Fig 1. A stepwise workflow was employed to exploring *Dolichos lablab* compounds as potential inhibitors for Fusion (F) protein of human metapneumovirus (HMPV).**

Autodock after the ligand and substrate had been prepared exactly. The designated grid box at the fusion protein docking site had the following coordinates: Dimensions (Angstrom) of X: 114.0830 Y: 58.6489 Z: 103.6633, with a center of X: 7.3656 Y: 3.7987 Z: 44.0685 Å. The conformation with the highest docking energy once molecular docking was complete represented the preeminent conformation. After docking, the selected compounds, along with the co-crystallized reference ligand NAG (2-acetamido-2-deoxy-beta-D-glucopyranose), were re-docked into the active site of the HMPV F protein using the Mcule 1-Click Docking platform to evaluate their binding affinity [27]. To ensure the reliability of the docking protocol, the root-mean-square deviation (RMSD) between the docked and crystallographic conformations of the reference ligand NAG was calculated using UCSF Chimera [28].

### Pharmacokinetics, drug-like properties, and toxicity profile analysis

The ADMET structure-activity relationship (admetSAR) [29], SwissADME [30], Deep-PK [31], and pKCSM [32] tools were employed as indispensable web-based servers to study and assess the physicochemical characteristics in conjunction

with the pharmacokinetic parameters. The medicinal chemistry compatibility of the selected, likely antiviral phytochemicals is predicted by the Canonical Simplified Molecular-Input Line-Entry System (SMILES), which is retrieved from the Pub-Chem database and utilized by the previously defined web services.

### Prediction of physicochemical properties related to drug-likeness of the drug candidates

By using an online cheminformatics platform, namely, Molinspiration [33] to predict the physicochemical properties related to drug-likeness of our lead compounds. Several physicochemical properties parameters were predicted using SMILES of our phytochemicals. This program uses advanced Bayesian statistics to assess a training set of active structure and compare it to inactive molecules [34].

### Studies of molecular dynamic simulation

To evaluate the binding stability of the three selected candidates, a 100 ns simulation was performed to investigate the protein-ligand complexes. Molecular dynamics simulations were conducted on Desmond Maestro 2020 systems with the OPLS4 force field operating on Linux to assess different protein–ligand complex structures [35]. Additionally, the TIP3P aqueous archetype was used to set up a predetermined volume with an orthorhombic periodic boundary box. The physiological conditions were set for the simulation cell, which comprised 310 K temperature, 0.15 M NaCl (sodium chloride), and pH 7.0. The protein-ligand solvated complex was then exposed to 100 ns of the energy minimization. The system was heated to 300K after all of the hydrogen atoms were eliminated using the SHAKE method [36]. 1.25 ns was used as the time step of the simulation. The simulation was prolonged up to 100 ns periods. The Root Mean Square Deviation (RMSD), Root Mean Square Fluctuation (RMSF), Radius of Gyration (Rg), Solvent-Accessible Surface Area (SASA), and intermolecular bonding were all estimated using trajectories [37–41]. Lastly, trajectories snapshots were taken at 100 ps intervals.

### Binding free energy calculation (MMGBSA)

The values of binding free energy were predicted through PRODIGY, a web-based server [42]. The total energy (G) between the ligand (compound) and receptor (protein) was calculated as:

$$\Delta G_{predicted} = 0.0115148 \times E_{elec} - 0.0014852 AC_{CC} + 0.0057097 \times AC_{NN} - 0.1301806 \times AC_{XX} - 5.1002233 \tag{1}$$

Where the electrostatic energy is denoted by $E_{elec}$ and the atomic contacts between carbon and carbon, nitrogen and nitrogen, and all other atoms and polar hydrogens are denoted by $AC_{CC}$, $AC_{NN}$, and $AC_{XX}$, respectively.

## Results and discussion

### Analysis of molecular docking

Molecular docking revealed that brassinolide (CID_115196), lanosterol (CID_246983), quercetin (CID_5280343), beta-carotene (CID_5280489), stigmasterol (CID_5280794), 2'-hydroxygenistein (CID_5282074), cholesterol (CID_5282074), gibberellin A4 (CID_92109), trans-zeatin glucoside (CID_5280489), psilostachyin B (CID_5320768), rutin (CID_5280805), isoquercetin (CID_5280804), ilicic acid (CID_496073), oleanolic acid (CID_10494), nandrolone (CID_9904), and ursolic acid (CID_64945) exhibited robust binding affinities with the HMPV F protein (Table 1). Docking poses of the final three candidates are displayed in (Fig 2).

To validate the molecular docking, a re-docking procedure was carried out for the 16 top-ranked compounds based on their docking scores, with the objective of evaluating their interaction with the target protein. The docking grid was precisely centered at coordinates X = 7.3656, Y = 3.7987, and Z = 44.0685, with each axis (X, Y, Z) extended by 20 Å to fully

**Table 1. Binding affinity between Fusion (F) protein of HMPV and Dolichos Lablab (Lead compounds).**

| Receptor | Compounds | Binding Affinity (Kcal/mol) |
|---|---|---|
| Fusion (F) protein of HMPV | Brassinolide | −8.2 |
| | Lanosterol | −7.9 |
| | Quercetin | −7.6 |
| | beta-Carotene | −9.7 |
| | Stigmasterol | −8.3 |
| | 2'-Hydroxygenistein | −7.6 |
| | Cholesterol | −7.6 |
| | Gibberellin A4 | −7.9 |
| | trans-Zeatin glucoside | −7.6 |
| | Psilostachyin B | −13.9 |
| | Rutin | −9.1 |
| | Isoquercetin | −7.9 |
| | Ilicic Acid | −8.9 |
| | Oleanolic Acid | −7.5 |
| | Nandrolone | −8.6 |
| | Ursolic Acid | −8.1 |
| | Ribavirin | −6.7 |

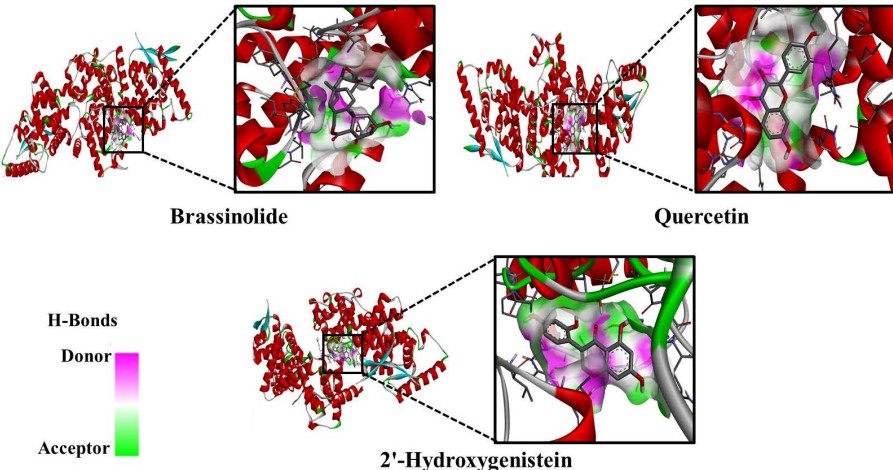

**Fig 2. Docking poses of Brassinolide, Quercetin, and 2'-Hydroxygenistein with Fusion (F) protein of HMPV.**

encompass the binding site. For validation of the docking protocol, a control complex from the Protein Data Bank (PDB) was employed, wherein the re-docked ligand demonstrated a RMSD of 0 Å relative to the co-crystallized ligand NAG. This perfect alignment confirms the ability of the docking method to accurately reproduce the experimentally determined binding conformation, thereby reinforcing the reliability of the approach. Furthermore, the re-docking analysis reaffirmed the favorable binding energies of 16 top-ranked compounds, which were subsequently selected for further investigation based on their promising interaction profiles. This validation step provided a critical benchmark for ensuring the consistency and robustness of the docking results.

Our investigation found that brassinolide interacted with the HMPV fusion protein through two conventional H-bonds at positions LYS254 and ASP336. Similarly, quercetin formed one conventional H-bond at LEU158; one unfavorable donor-donor bond at ARG156; one unfavorable acceptor-acceptor bond at THR45; one pi-sigma bond at VAL148; two pi-pi stacked bonds at TYR44; and three pi-alkyl bonds at ARG156 and PRO235. 2'-hydroxygenistein formed three conventional H-bonds at TYR44, THR45, and ARG156; three pi-alkyl bonds at VAL148 and ARG156; one pi-sigma bond at VAL148; and two pi-pi stacked bonds at TYR44 on the active site of the target protein. Non-bonding interactions between the fusion protein of HMPV and the final three compounds in ([Fig 3]). The majority of the interactions were localized within regions associated with the heptad repeat domains (HR1 and HR2) and the fusion peptide, which are essential for membrane fusion and viral entry. These regions have also been implicated in the binding of neutralizing antibodies, supporting the potential functional relevance of the identified docking sites.

The binding affinity of ribavirin (CID_37542) was −6.7 kcal/mol as a control following docking with the fusion protein of HMPV using the prepared grid, which was significantly lower than the binding affinity of these leading compounds. Ribavirin is a broad-spectrum antiviral compound that reduces RNA-dependent RNA polymerase activity. Primary mechanism of

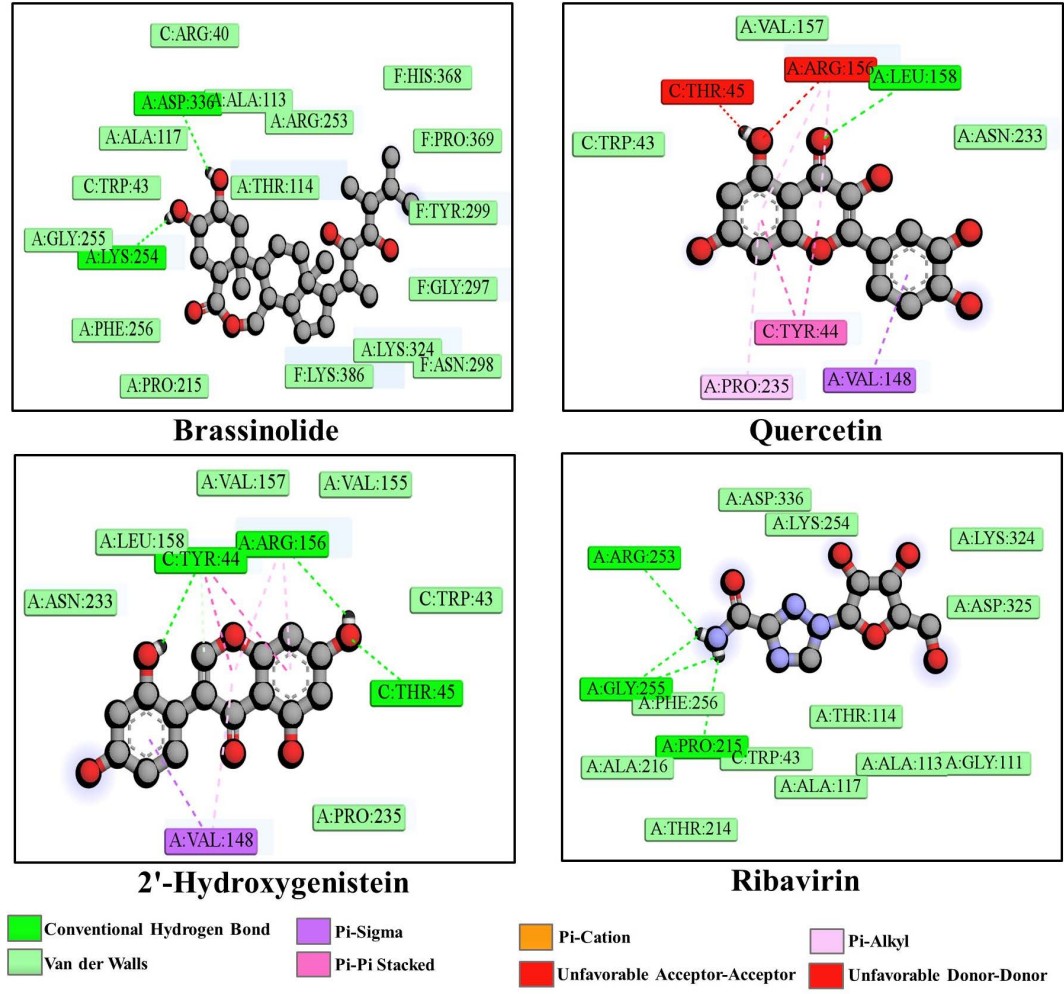

**Fig 3. Non-bonding interactions between the fusion protein of HMPV and Brassinolide, Quercetin, 2'-Hydroxygenistein, and Ribavirin.**

action of ribavirin involves inhibition of the viral RNA-dependent RNA polymerase, not the F protein. In our study, ribavirin was employed as a reference antiviral agent due to its reported activity against HMPV in previous studies, not to imply a direct interaction with the F protein. It has shown in vitro activity against HMPV, but its clinical use is limited due to potential toxicity and lack of definitive efficacy in vivo [43,44].

## Analysis of pharmacokinetics, drug-like properties, and toxicity profile

The pharmacological activity and safety of brassinolide, lanosterol, quercetin, beta-carotene, stigmasterol, 2'-hydroxygenistein, cholesterol, gibberellin A4, trans-zeatin glucoside, psilostachyin B, rutin, isoquercetin, ilicic acid, oleanolic acid, nandrolone, and ursolic acid were evaluated by determining their drug-likeness characteristics (Table 2).

The drug-likeness of these *DL*-derived lead compounds was then analyzed using Lipinski's rule of 5. In this case, five [5] compounds with ribavirin, like brassinolide, quercetin, beta-carotene, 2'-hydroxygenistein, gibberellin A4, psilostachyin B, ilicic acid, and nandrolone filled 5 of Lipinski's rules with no violation. Compounds that violate one or more of Lipinski's criteria may face challenges in oral bioavailability and drug development. One of the most crucial factors in assessing a chemical's antiviral efficacy is its molecular weight. In contrast to large molecular weight molecules, molecules with a molecular weight of less than 500 g/mol are quickly transported, distributed, and absorbed by the cell membrane [45]. All selected compounds, with the exception of beta-carotene and rutin, exhibited molecular weights below 500 g/mol, consistent with the threshold commonly associated with favorable drug-likeness. Additionally, chemicals can pass through the cell membrane more easily when the MlogP values are positive; a value of less than five is acceptable [46,47]. By passive diffusion, the lipophilic chemicals readily penetrate the cell membrane and bind with molecules as inhibitors. Consequently, the lipophilic nature of the chemical determines the membrane permeability. Among the evaluated compounds, brassinolide, quercetin, beta-carotene, 2'-hydroxygenistein, gibberellin A4, psilostachyin B, rutin, isoquercetin, ilicic acid, nandrolone, and ribavirin are ideal for penetrating the cell membrane. According to a recent study, the mono-alkyl lipophilic cation C18-SMe2+, which has an MlogP value of 2.26, diffuses easily through the plasma membrane [48].

**Table 2.** Drug-likeness properties of lead compounds using SwissADME.

| Compounds | Molecular Weight | MLogP | H-bond acceptor | H-bond donor | Lipinski |
|---|---|---|---|---|---|
| Brassinolide | 480.68 | 3.05 | 6 | 4 | Yes; 0 violation |
| Lanosterol | 426.72 | 6.82 | 1 | 1 | Yes; 1 violation: MLOGP>4.15 |
| Quercetin | 302.24 | −0.56 | 7 | 5 | Yes; 0 violation |
| beta-Carotene | 536.87 | 2.56 | 0 | 0 | Yes; 0 violation |
| Stigmasterol | 412.69 | 6.62 | 1 | 1 | Yes; 1 violation: MLOGP>4.15 |
| 2'-Hydroxygenistein | 286.24 | −0.03 | 6 | 4 | Yes; 0 violation |
| Cholesterol | 386.65 | 6.34 | 1 | 1 | Yes; 1 violation: MLOGP>4.15 |
| Gibberellin A4 | 332.39 | 2.56 | 5 | 2 | Yes; 0 violation |
| trans-Zeatin glucoside | 381.38 | −2.72 | 11 | 6 | No; 2 violations: NorO>10, NHorOH>5 |
| Psilostachyin B | 262.30 | 2.34 | 4 | 0 | Yes; 0 violation |
| Rutin | 610.52 | −3.89 | 16 | 10 | No; 3 violations: MW>500, NorO>10, NHorOH>5 |
| Isoquercetin | 464.38 | −2.59 | 12 | 8 | No; 2 violations: NorO>10, NHorOH>5 |
| Ilicic Acid | 252.35 | 2.56 | 3 | 2 | Yes; 0 violation |
| Oleanolic Acid | 456.70 | 5.82 | 3 | 2 | Yes; 1 violation: MLOGP>4.15 |
| Nandrolone | 274.40 | 3.36 | 2 | 1 | Yes; 0 violation |
| Ursolic Acid | 456.70 | 5.82 | 3 | 2 | Yes; 1 violation: MLOGP>4.15 |
| Ribavirin | 244.206 | −1.85 | 7 | 4 | Yes; 0 violation |

Moreover, an efficient drug candidate has less than 5 hydrogen bond donors and less than 10 hydrogen bond acceptors [49]. In this case, brassinolide, lanosterol, quercetin, beta-carotene, stigmasterol, 2'-hydroxygenistein, cholesterol, gibberellin A4, psilostachyin B, ilicic acid, nandrolone, and ribavirin showed less than 5 hydrogen bond donors and 10 hydrogen bond acceptors. In our study, the molecular weight and MLogP value of brassinolide, quercetin, beta-carotene, 2'-hydroxygenistein, gibberellin A4, psilostachyin B, ilicic Acid, and nandrolone exceeded the anticipated limit mentioned in the Lipinski's rule of 5.

The central nervous system (CNS) permeability, p-glycoprotein inhibition, cytochrome P450 (CYP) inhibition, carcinogenicity, and hepatotoxicity of these phytochemicals were evaluated as well. The ability of a substance to cross the selectively semipermeable blood-brain barrier is known as CNS permeability in this context [50]. The central nervous system can only be penetrated if the permeability value of the CNS is higher than −2, according to research [51]. Our lead phytochemicals evaluated as permeability values of CNS are higher than −2, except beta-carotene (−1.074), stigmasterol (−1.652), cholesterol (−1.75), oleanolic acid (−1.176), and ursolic acid (−1.187), as well as brassinolide, 2'-hydroxygenistein, rutin, and isoquercetin are blood-brain barrier non-penetrable (high confidence); trans-zeatin glucoside is non-penetrable (low confidence); and the other 10 compounds with ribavirin are penetrable (high confidence). These phytochemicals also did not exhibit hepatotoxicity or acute oral toxicity except for gibberellin A4, oleanolic acid, and ursolic acid. Clearance of drug range: low clearance (<10 mL/min/kg), moderate clearance (10–50 mL/min/kg), and high clearance (>50 mL/min/kg). The clearance range of cholesterol, rutin, isoquercetin, and nandrolone is 13.16, 13.30, 13.22, and 17.09, which means moderate clearance, and 12 other lead compounds with ribavirin showed less than 10 mL/min/kg, which means low clearance (Table 3) and detailed in S2 Table.

## Prediction of the physicochemical properties related to drug-likeness of the drug candidates

To evaluate the physicochemical properties related to drug-likeness of lead compounds with high potential, a number of observations required careful analysis. Physicochemical properties related to drug-likeness parameters of drug candidates like topological polar surface area (TPSA), volume, and number of rotatable bonds (nrotb) and MLogP. The topological polar surface area (TPSA) of a drug is typically less than or equal to 140 Å$^2$. When TPSA ≤ 140 Å$^2$, the drug candidate

**Table 3. Pharmacokinetics properties of selected lead five compounds.**

| Properties | Brassinolide | Quercetin | 2'-Hydroxygenistein | Rutin | Isoquercetin | Ribavirin |
|---|---|---|---|---|---|---|
| CNS Permeability (LogPS) | −3.115 | −3.065 | −2.394 | −5.178 | −4.093 | −1.256 |
| CYP2D6 substrate | No | No | No | No | No | No |
| CYP3A4 substrate | Yes | No | Yes | No | No | No |
| CYP1A2 inhibitor | No | Yes | Yes | No | No | No |
| CYP2C19 inhibitor | No | No | No | No | No | Yes |
| CYP2C9 inhibitor | No | No | No | No | No | Yes |
| CYP2D6 inhibitor | No | No | No | No | No | No |
| CYP3A4 inhibitor | No | No | Yes | No | No | No |
| Ames Toxicity | No | No | No | No | No | Yes |
| Hepatotoxicity | No | No | No | No | No | No |
| Acute Oral Toxicity (log(1/(mol/kg)) | 2.777 | 2.471 | 2.291 | 2.491 | 2.541 | Yes |
| Bioavailability Score | 0.55 | 0.55 | 0.55 | 0.55 | 0.17 | 0.55 |
| Blood-Brain Barrier (BBB) | Non-Penetrable (High Confidence) | Non-Penetrable (High Confidence) | Non-Penetrable (High Confidence) | Non-Penetrable (High Confidence) | Non-Penetrable (High Confidence) | Penetrable (High Confidence |
| Skin Sensitisation | No | No | No | No | No | No |
| Clearance | 4.30 | 8.91 | 5.42 | 13.30 | 13.22 | 6.52 |

has good oral bioavailability and efficient transfer inside the intestine and BBB. When TPSA > 140 Å², drug has poorly absorbed [52]. In this analysis, rutin, isoquercetin, trans-zeatin glucoside, and ribavirin exhibited the highest topological polar surface area (TPSA) values, which are indicative of poor intestinal absorption. In contrast, the remaining 13 compounds demonstrated relatively low TPSA values, suggesting favorable oral bioavailability... Molecular volume ranges from 100 to 500 Å³, indicating small molecules, and molecular volume ranges from >500 Å³, indicating larger drug molecules [53]. Here, without beta-carotene (591.96), 15 other compounds have less than 500 Å³ molecular volume. Number of Rotatable Bonds (nrotb) evaluated as low Flexibility (nrotb ≤ 5), which is common in small, rigid molecules with good oral bioavailability, moderate Flexibility (5 < nrotb ≤ 10), and high Flexibility (nrotb > 10) [54]. According to the number of rotatable bonds, brassinolide, lanosterol, quercetin, stigmasterol, 2'-hydroxygenistein, cholesterol, gibberellin A4, psilostachyin B, isoquercetin, ilicic acid, oleanolic acid, nandrolone, ursolic acid, and ribavirin evaluated as nrotb ≤ 5, which means low flexibility; beta-carotene, trans-zeatin glucoside, and rutin evaluated as 5 < nrotb ≤ 10, which means moderate flexibility. Detailed in Table 4.

According to physicochemical properties related to drug-likeness parameters of drug candidates, brassinolide, lanosterol, quercetin, beta-carotene, stigmasterol, 2'-hydroxygenistein, cholesterol, gibberellin A4, psilostachyin B, ilicic acid, oleanolic acid, nandrolone, and ursolic acid showed preeminent TPSA, molecular volume (Å³), and Number of rotatable bond (nrotb) and may biologically active compounds. Following a comprehensive analysis of pharmacokinetics, drug-likeness, toxicity profiles, and physicochemical properties related to drug-likeness, Brassinolide, Quercetin, and 2'-Hydroxygenistein were identified as promising drug candidates. Chemical scaffold of the final three DL-derived compounds is shown in Fig 4. It is important to note that these findings are computational predictions and require extensive experimental validation to confirm their bioavailability and therapeutic potential.

## Molecular dynamics simulation study

Molecular dynamics simulation runs on a real-time phase to demonstrate the protein–ligand complex stability in a controlled environment similar to the human body [55]. Additionally, it provides data about the change of protein complex

Table 4. Predicted physicochemical properties related to drug-likeness of lead compounds.

| Compounds | TPSA | Molecular Volume (Å³) | Number of Rotatable Bonds (nrotb) |
|---|---|---|---|
| Brassinolide | 107.22 | 481.23 | 5 |
| Lanosterol | 20.23 | 465.95 | 4 |
| Quercetin | 131.35 | 240.08 | 1 |
| beta-Carotene | 0.00 | 591.96 | 10 |
| Stigmasterol | 20.23 | 450.33 | 5 |
| 2'-Hydroxygenistein | 111.12 | 232.07 | 1 |
| Cholesterol | 20.23 | 423.13 | 5 |
| Gibberellin A4 | 83.83 | 300.17 | 1 |
| trans-Zeatin glucoside | 166.01 | 330.09 | 6 |
| Psilostachyin B | 52.61 | 241.68 | 0 |
| Rutin | 269.43 | 496.07 | 6 |
| Isoquercetin | 210.50 | 372.21 | 4 |
| Ilicic Acid | 57.53 | 254.30 | 2 |
| Oleanolic Acid | 57.53 | 471.14 | 1 |
| Nandrolone | 37.30 | 275.30 | 0 |
| Ursolic Acid | 57.53 | 471.49 | 1 |
| Ribavirin | 143.72 | 203.5 | 3 |

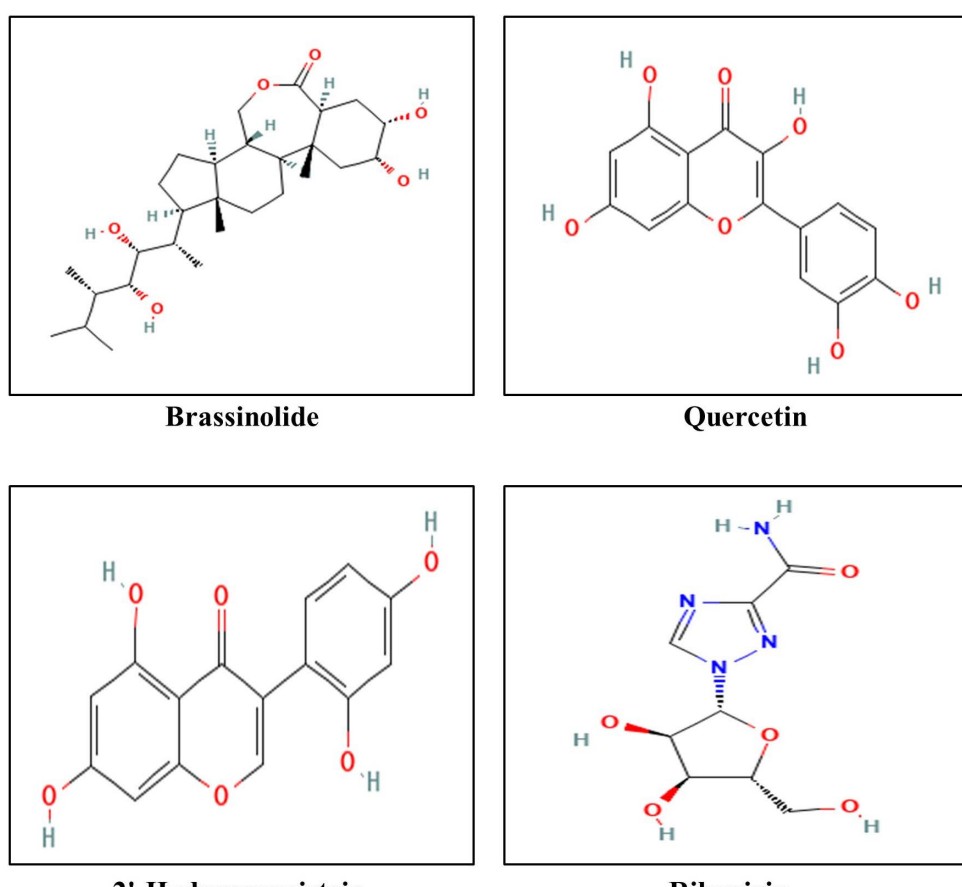

**Brassinolide**

**Quercetin**

**2'-Hydroxygenistein**

**Ribavirin**

**Fig 4. Chemical scaffold of Brassinolide, Quercetin, 2'-Hydroxygenistein, and Ribavirin.**

conformation in computational systems. For the best justification of complex stability, the selected four protein–compound complexes along with the protein–reference complex was subjected to a 100 ns simulation to find out the most stable compounds in this assay.

## Root mean square deviation (RMSD)

Root Mean Square Deviation (RMSD) quantifies the deviation of protein structures from a reference conformation throughout MD simulations. Protein–ligand interactions with an average RMSD value change of 1–3 Å is an acceptable range for MD simulation [56]. If the value crosses the average range, then the protein structure may go through a conformational change during interactions with ligands. Analyzing the RMSD results, compared to the ribavirin_7SEJ complex, the stability of three selected complexes remained quite stable throughout the simulation, indicating fewer structural deviations [57,58]. (Fig 5a).

## Root mean square fluctuation (RMSF)

Similar to RMSD, Root Mean Square Fluctuation (RMSF) is a numerical metric that determines how much a particular residue fluctuates over the duration of a simulation rather than showing positional variations over time between whole structures [59]. The RMSF also revealed insights on the flexibility of each atom in the ligands [60]. The changes that

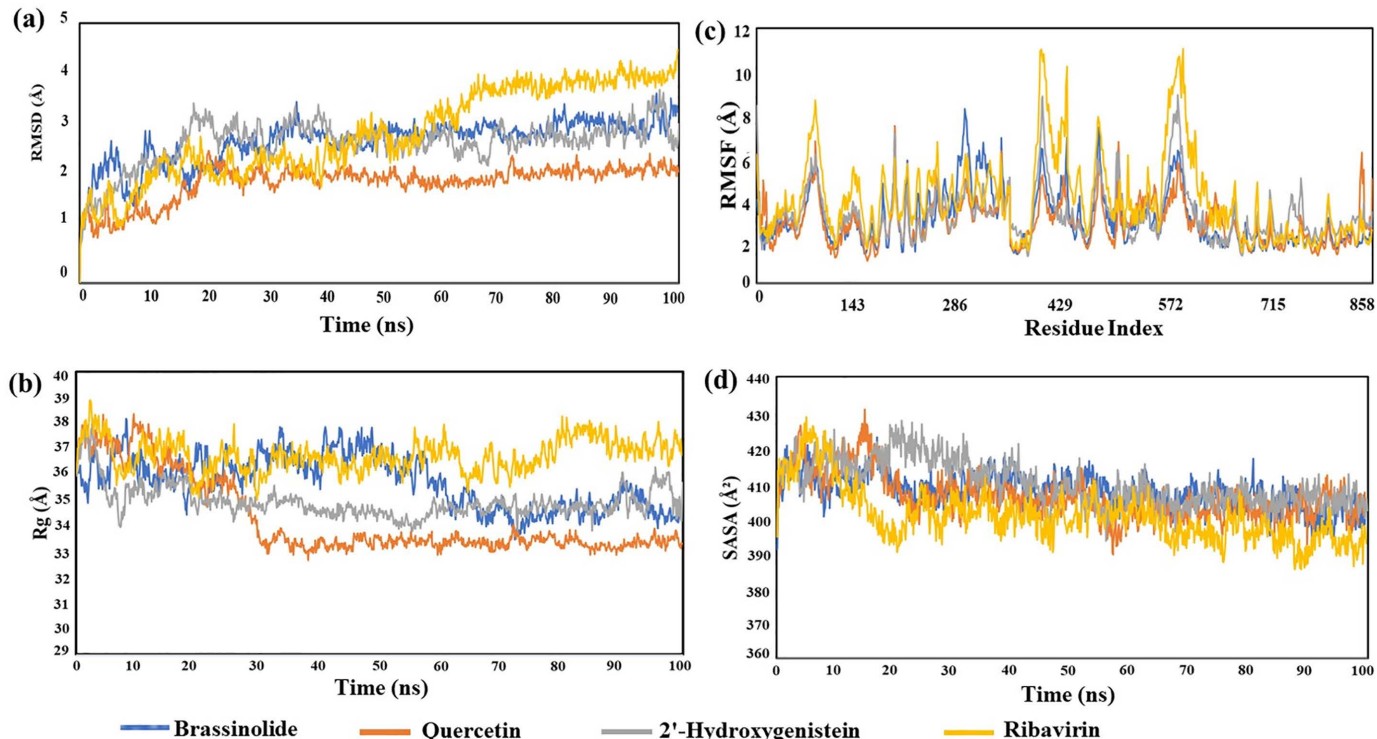

**Fig 5. Molecular dynamics simulation of three selected protein–ligand complexes and a ribavirin compound with a 100 ns runtime. (a)** Root Mean Square Deviation (RMSD); **(b)** Root Mean Square Fluctuation (RMSF); **(c)** Radius of Gyration (Rg); and **(d)** Solvent accessible surface area (SASA).

occur within the amino acid residues of the protein chain during protein–ligand interactions are mainly determined by RMSF. In this research, the RMSF values of brassinolide_7SEJ, quercetin_7SEJ, and 2'-hydroxygenistein_7SEJ, and the ribavirin_7SEJ model were calculated to detect the changes of protein structure and amino acid composition caused by small molecules attaching to a particular target protein and its residues. The RMSF values for the brassinolide_7SEJ, quercetin_7SEJ, and 2'-hydroxygenistein_7SEJ, and ribavirin_7SEJ complexes were 3.047, 2.860, 3.227, and 3.951 Å, respectively. Compared to the ribavirin_7SEJ complex, the RMSF value for the three selected complexes was less fluctuating, which indicated their lower flexibility, greater stability and rigidity compared to the ribavirin_7SEJ complex (Fig 5b).

## Radius of gyration (Rg)

In the model of interactions between protein and small molecules, the configuration of atoms along its axis is ascertained via the investigation of the radius of gyration (Rg). Rg is the most valuable prediction model because it helps to provide the calculation and conception of the compactness of the entire complex during simulation period [61]. Thus, it helps clearly to see the possibility of macromolecule structural feasibility. The compound ribavirin_7SEJ complex showed the greatest Rg values, suggesting a more stretched shape and a wider dispersion of atoms from the center of mass. Conversely, the complexes of brassinolide_7SEJ, quercetin_7SEJ, and 2'-hydroxygenistein_7SEJ showed the lowest RG value, indicating a more rigid and stable structure throughout the protein-ligand structure (Fig 5c).

## Solvent accessible surface area (SASA)

Solvent accessible surface area (SASA) is a great indication of protein folding and stability [62]. SASA is a crucial metric for assessing the stability and folding of proteins since higher SASA values indicate a larger protein surface area, while lower SASA values indicate a smaller protein surface area [63]. Target protein surface areas contain specific amino acid residues that small molecule ligands interact with through hydrophilic or hydrophobic interactions, the values of which can be ascertained using SASA, as hydrophobic amino acids may be one of the reasons for protein folding. Our research demonstrated that the quercetin_7SEJ and 2'-hydroxygenistein_7SEJ complexes had lower SASA (392.59 Å$^2$ and 401.87 Å$^2$), indicating more of the surface of quercetin. On the other hand, brassinolide_7SEJ complex exhibited higher SASA (404.94 Å$^2$) than ribavirin_7SEJ complex, indicates a larger portion of the Brassinolide is exposed to the solvent (water), which can weaken interactions with the Fusion protein and potentially decrease the brassinolide's potency (Fig 5d).

## Intermolecular bonds

Using a simulation duration of 100 ns, the intermolecular bonds of protein–ligand complexes were evaluated. Water bridges, ionic bonds, hydrogen bonds, and hydrophobic interactions are represented in Fig 6. For the brassinolide_7SEJ complex (Fig 6a), 11 hydrogen bonds were discovered for a short period; among them, 3 significant hydrogen bond interaction was visualized at F:SER371 (45%), F:VAL373 (65%), and F:TYR425 (60%). 3 Hydrophobic bonds were also occupied. Besides them, 22 water bridges were observed. Among them, F:SER371 demonstrated for 50% simulation time period. In the case of quercetin_7SEJ complex (Fig 6b), 5 hydrogen bonds were observed. Among them, C:THR45 demonstrated as significant (75%). 4 hydrophobic bonds were spotted as well as 8 water bridges were also observed. Moving to the 2'-hydroxygenistein_7SEJ complex (Fig 6c), 12 hydrogen bonds were from which hydrogen bonds of A:ASP325 (75%) and A:ASP336 (45%) was notable. Only one hydrophobic bonds at A:LYS254 was also observed. 18 water bridges were also be found, from them A:LYS254 demonstrated for 40% simulation time period formed. the In contrast to the selected compounds, the ribavirin_7SEJ complex (Fig 6d), 5 hydrogen bonds were found at A:ARG156, A:ASN233, C:TRP43, C:TYR44, and C:THR45, from which hydrogen bond of C:THR45 (60%) was notable. 5 hydrophobic bonds at A:VAL148, A:ARG156, A:LEU158, A:PRO235, and C:TYR44 were also observed. 7 water bridges were also be, from them A:ARG156 demonstrated for 50% simulation time period formed, which proves that our selected compounds are far better than the ribavirin.

## Binding free energy (MMGBSA)

Ligand binding with the receptor was further confirmed by analysing MMGBSA binding free energy calculations. The binding affinities of three selected complexes were assessed using the PRODIGY server, which demonstrated negative values, indicating robust binding and stability within the binding pocket. Analysis of the average binding free energy values revealed that all selected compounds displayed higher binding affinities compared to the ribavirin compound (Fig 7). Notably, quercetin exhibited the highest binding score among the selected compounds. This observation was confirmed by the stable profiles of RMSD, Rg, and SASA of the complexes.

After all analysis, we found 3 selected compounds exhibited higher binding affinities compared to the ribavirin compound, indicating constant interactions with the target protein. This study concludes by highlighting the potential of bioactive compounds derived from *Dolichos lablab* as HMPV fusion protein inhibitors. These findings highlight the significance of natural bioactive compounds in drug discovery and development by suggesting that brassinolide, quercetin, and 2'-hydroxygenistein are promising potential inhibitory and drug candidates that require further in vitro investigation.

## Conclusion

In this research, we identified potential or hypothetical inhibitors against the HMPV F protein that causes acute respiratory infections using several computational methods. To find the most potential or hypothetical lead compounds, the phytochemical library obtained from *DL* was investigated utilizing molecular docking against the HMPV target (F protein).

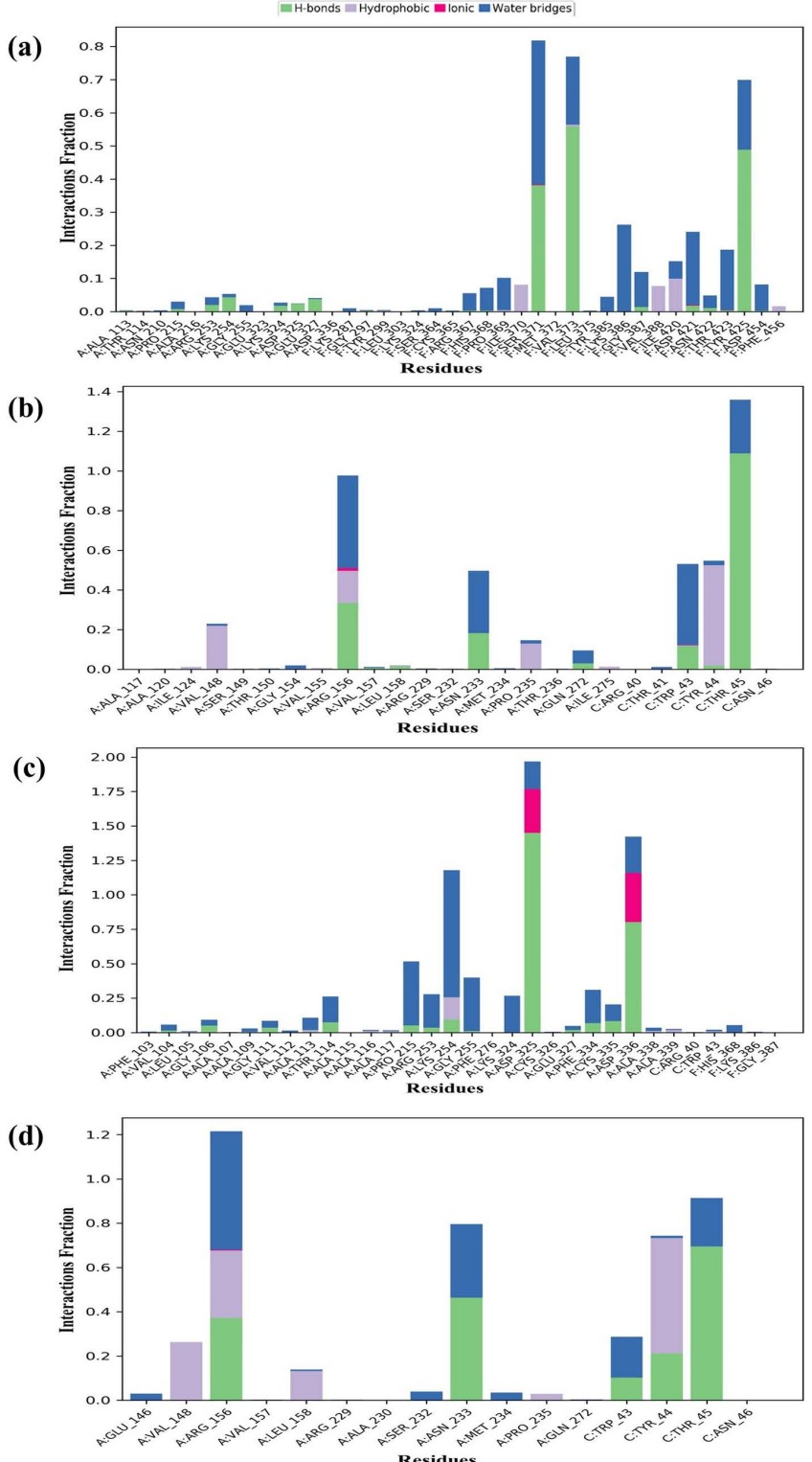

**Fig 6. Protein–ligand interactions through various types of bonds at 100 ns simulation running time.** The selected compounds Brassinolide, Quercetin, 2'-Hydroxygenistein and ribavirin complexed with the target protein were marked as a, b, c, and d respectively.

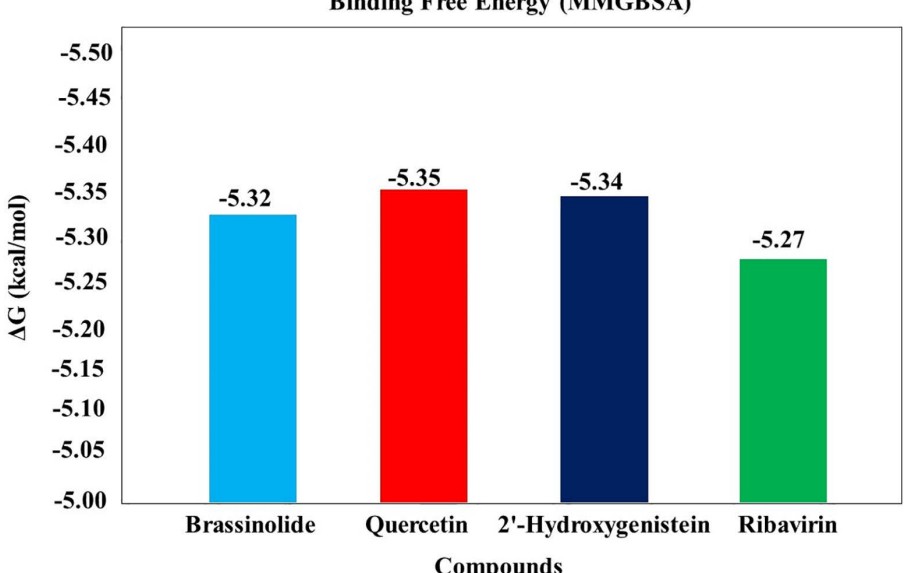

**Fig 7. Binding free energy values for the top three protein–ligand complexes and one ribavirin complex obtained from the PRODIGY server have been visualized in the graph.**

Additionally, it has been found that the HMPV F protein binds strongly to the top ligand molecules in the library, which include brassinolide, lanosterol, quercetin, beta-carotene, stigmasterol, 2'-hydroxygenistein, cholesterol, gibberellin A4, trans-zeatin glucoside, psilostachyin B, rutin, isoquercetin, ilicic acid, oleanolic acid, nandrolone, and ursolic acid. For the docked protein-ligand complexes, molecular dynamic simulation was also used to determine the stiffness and binding orientation. Simulation descriptors like RMSD, RMSF, RG, and SASA, as well as hydrogen bond descriptors, helped to analyse the rigid nature of the complexes in an atomistic setting. The drug-like characteristics, toxicity, and carcinogenicity of these top-ranked compounds were thoroughly studied using several computational approaches, and no harmful and unfavorable consequences have been observed.

While these findings provide valuable insights, it is important to acknowledge the limitations of the study. The predicted inhibitory effects have not yet been experimentally validated, as the conclusions are solely based on computational analyses. Furthermore, given the potential of brassinolide, quercetin, and 2'-hydroxygenistein as therapeutic agents, it is essential to thoroughly evaluate their pharmacokinetic, toxicological, and safety profiles. An additional limitation of this study is the exclusive use of Ribavirin as the reference compound in molecular docking, molecular dynamics (MD) simulations, and MM/GBSA analyses. The absence of additional controls, such as a known non-binder or a randomly selected ligand with no expected affinity for the HMPV F protein, limits the comparative robustness of the analysis.

To substantiate the computational predictions, experimental validation using cellular and in vivo models should be prioritized in future research. Comprehensive investigations into the pharmacokinetics, toxicity, and potential off-target effects are essential to establish the safety and therapeutic viability of these compounds. Structural optimization may further enhance their drug-like properties, binding affinity, and target selectivity. Additionally, exploring their activity against human metapneumovirus could potentially broaden their antiviral applications.

The development of *Dolichos lablab*-derived HMPV inhibitors can be accelerated through the integration of advanced computational techniques, including machine learning-based approaches, with rigorous experimental validation. These approaches offer novel strategies for antiviral drug discovery, potentially enhancing the therapeutic efficacy of brassinolide, quercetin, and 2'-hydroxygenistein, while contributing to preparedness for current and emerging pandemics.

## Supporting information

**S1 Table. Compound Name and PubChem CID of *Dolichos lablab*.**
(DOCX)

**S2 Table. Pharmacokinetics properties of lead compounds.**
(DOCX)

## Acknowledgments

None.

## Author contributions

**Conceptualization:** Md. Mainuddin Hossain, Md. Tanzimur Rahman Tanjil, Md. Mahbub Morshed.

**Data curation:** Md. Mainuddin Hossain, Md. Jahid Hasan Apu, Antora Kar.

**Formal analysis:** Md. Mainuddin Hossain, Md. Tanzimur Rahman Tanjil, Liton Chandra Das, Antora Kar.

**Methodology:** Md. Mainuddin Hossain.

**Software:** Md. Mainuddin Hossain, Md. Jahid Hasan Apu, Liton Chandra Das.

**Supervision:** Fatematuz Zuhura Evamoni, Md. Mahbub Morshed.

**Writing – original draft:** Md. Mainuddin Hossain, Md. Mahbub Morshed.

**Writing – review & editing:** Md. Faisal Bin Abdul Aziz, Fatematuz Zuhura Evamoni.

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
