## [Decision Letter · Decision Letter 0]

16 Jul 2025

PONE-D-25-32598Exploring Dolichos lablab Compounds as Potential Inhibitors for Fusion (F) Protein of Human Metapneumovirus (HMPV): A Systematic Computational ApproachPLOS ONE

Dear Dr. Morshed,

Thank you for submitting your manuscript to PLOS ONE. After careful consideration, we feel that it has merit but does not fully meet PLOS ONE’s publication criteria as it currently stands. Therefore, we invite you to submit a revised version of the manuscript that addresses the points raised during the review process.

**ACADEMIC EDITOR: **The submission reflects scientific relevance. However, some fundamental issues limit its quality for publication in the current form. For instance, the authors need to justify the significance of the study, relate it to the literature and identify the gap in the existing knowledge that this aims to satisfy, and ensure an adequate validation of the theoretical analysis. Again, what are the limitations of this study and how can the authors recommend future research on the study. Moreover, some concerns have been raised by the reviewers affecting certain sections of the study. Kindly pay a thorough attention to these and address them critically before resubmission.

We look forward to receiving your revised manuscript.

Kind regards,

Yusuf Oloruntoyin Ayipo, Ph.D

Academic Editor

PLOS ONE

Journal Requirements:

3. Please remove all personal information, ensure that the data shared are in accordance with participant consent, and re-upload a fully anonymized data set.

Additional Editor Comments :

The submission reflects scientific relevance. However, some fundamental issues limit its quality for publication in the current form. For instance, the authors need to justify the significance of the study, relate it to the literature and identify the gap in the existing knowledge that this aims to satisfy, and ensure an adequate validation of the theoretical analysis. Again, what are the limitations of this study and how can the authors recommend future research on the study. Moreover, some concerns have been raised by the reviewers affecting certain sections of the study. Kindly pay a thorough attention to these and address them critically before resubmission.

Reviewers' comments:

Reviewer's Responses to Questions

**Comments to the Author**

1. Is the manuscript technically sound, and do the data support the conclusions?

Reviewer #1: Yes

Reviewer #2: Yes

Reviewer #3: Yes

2. Has the statistical analysis been performed appropriately and rigorously? 

Reviewer #1: Yes

Reviewer #2: Yes

Reviewer #3: Yes

3. Have the authors made all data underlying the findings in their manuscript fully available?

Reviewer #1: Yes

Reviewer #2: Yes

Reviewer #3: Yes

4. Is the manuscript presented in an intelligible fashion and written in standard English?

Reviewer #1: Yes

Reviewer #2: Yes

Reviewer #3: Yes

5. Review Comments to the Author

Reviewer #1: Authors report a systematic computational approach to screening compounds derived from a plant (Dolichos lablab) in search of an effective and safe antiviral capable of targeting the fusion protein of the human metapneumovirus to combat prevalence of resulting respiratory infections.

Manuscript requires a couple of corrections before acceptance:

First 2 paragraphs (particularly the first) of the introduction are missing key information and need reorganization. A good way to introduce is properly address respiratory infections and its burden, then viruses/ respiratory viruses. Also, we cannot address the issue of respiratory viruses without mentioning the likes of Influenza, SarsCoV2, and RSV as main causes.

In the sentence “Viruses have enzymes (polymerases) that help them with genome replication….”, authors should note that not all viruses have and/or use polymerases.

In the intro, kindly use hMPV or HMPV, not both.

In the sentence “Natural products, also referred to as plant derived phytochemical…”, authors should note that natural products encompass those obtained from microbes and animals too, not only plants. Therefore, “also referred to as” can’t be used in the description.

“…phytochemicals that prevent viral reproduction and penetration…” correction: phytochemicals that prevent viral entry and replication.

It is known that HMPV F protein has the pre-Fusion and post-Fusion confirmation, with the former usually targeted for antivirals. Authors should clearly state in the intro or method that pre-fusion conformation was used.

Results section clearly stated how the 86 compounds of DL were reduced to 16 after analysis. Authors however, did not clearly mention how and why the 3 selected compounds were preferred for some of the analyses done. For example, authors commended that the 3 selected compounds have higher binding affinity than Ribavirin, but there were compounds within the list that had much better binding affinity and weren’t selected.

Authors should be consistent with keywords. Any of (a) or (b) should have the same keywords throughout the manuscript except if properly replaced first.

(a) Table 1 - Ligands; Table 2 - Compound name; Table 3 - Compounds

(b) control, Ribavirin, Ribavirin (control).

It will be helpful if authors include Ribavirin as a control in all their analyses including in the tables shown for comparison.

“…but its clinical use is limited due to potential toxicity and lack of definitive efficacy in vivo.” Please include appropriate citation here.

Rewrite format of references 4 and 24.

References 48 and 52 are not related to the statements they are cited with. Kindly review source of information.

Figure 6C is missing the y-axis title.

Figure 6ABCD are missing the x-axis title.

Figure 7 is missing x-axis title.

Authors should note that the mode of action/ target of Ribavirin is not the F protein, but the viral polymerase. Or is the author suggesting that Ribavirin potentially targets F protein of HMPV too? This doesn’t affect the the analyses done with Ribavirin as a control.

Very important; Revise the quality of all the figures and ensure they meet submission standards.

Reviewer #2: Thank you for giving me the opportunity to review this manuscript titled "Exploring Dolichos lablab Compounds as Potential Inhibitors for Fusion (F) Protein of Human Metapneumovirus (HMPV): A Systematic Computational Approach."

Summary and Strengths

This study presents a computational pipeline to identify and evaluate potential phytochemical inhibitors from Dolichos lablab targeting the HMPV fusion (F) protein. The authors integrate molecular docking, ADMET profiling, bioactivity prediction, and molecular dynamics simulations to identify promising drug candidates—Brassinolide, Quercetin, and 2'-Hydroxygenistein.

Specific strengths include:

Clear Target Rationale: The manuscript justifies the selection of the HMPV fusion protein as a therapeutic target, citing its role in viral entry and lack of existing effective inhibitors.

Use of Natural Product Library: The use of Dolichos lablab, a plant with traditional medicinal uses and known pharmacological activity, strengthens the phytochemical selection rationale.

Comprehensive Methodology:

Docking was conducted with AutoDock Vina and PyRx, with thoughtful grid definition.

ADMET and Lipinski analyses were thorough, using multiple tools (SwissADME, admetSAR, pKCSM, Deep-PK).

MD simulations were performed for 100 ns using Desmond with detailed reporting of RMSD, RMSF, Rg, SASA, and bonding interactions.

The inclusion of MMGBSA free energy calculations via PRODIGY supports the thermodynamic favorability of binding.

Comparison to a Known Control (Ribavirin): The use of Ribavirin as a benchmark adds a point of reference, improving interpretability.

Limitations and Concerns

Despite its strengths, there are several critical limitations that must be addressed before publication:

1. Over-interpretation of Computational Data

No experimental validation: All conclusions are based solely on in silico data. While the manuscript acknowledges this in the discussion, the conclusions and abstract use definitive language (e.g., “effective inhibitor,” “demonstrated promising biological activity”) that overstates the evidence.

Suggested fix: Tone down the certainty in both the abstract and conclusion, making it clear these are hypothetical inhibitors pending in vitro validation.

2. Redundancy and Clarity

The manuscript is excessively repetitive. For instance, the results sections discussing RMSD, RMSF, and SASA repeat similar information already presented earlier. Much of this can be consolidated to improve readability.

There are also long sections describing every value in a table, which can be summarized more efficiently (e.g., in bullet format or comparative sentences).

3. Misplaced Emphasis in the Introduction

The introduction spends an inordinate amount of time reviewing unrelated viruses (e.g., monkeypox, hantavirus), which dilutes focus from HMPV.

Suggested fix: Remove extraneous background on unrelated viruses and expand the part discussing existing drug discovery efforts for HMPV or other respiratory paramyxoviruses.

4. Scientific Rigor in Docking Analysis

Docking poses and interactions are described, but the manuscript does not discuss if the docking site corresponds to a functionally important site on the F protein. Do the identified binding sites overlap with known neutralizing epitopes or fusion machinery?

Suggestion: Provide structural or functional context for the binding sites—are they in the prefusion region, HR1/HR2 domains, or fusion loop?

5. Lack of Controls and Benchmarking

While Ribavirin is used as a control for docking energy, there is no control for the MD simulations or MMGBSA scoring other than that. Including a known non-binder or random ligand could contextualize the stability of the selected hits better.

6. Clarity of Figures and Tables

Figures (e.g., Fig 2, 3, 6) are referenced in the text but not included in the version I reviewed. Ensure all visual elements are clear, labeled, and aligned with journal standards.

There is inconsistency in compound naming (e.g., CID_5282074 used twice for different compounds; possible typo).

Recommendation: Major Revisions

Given the extensive and well-structured computational approach, the manuscript presents interesting preliminary findings. However, I recommend major revisions before this work can be considered for publication. Specifically:

Revise the abstract and conclusion to avoid overstatements.

Reduce redundancy and improve clarity in the writing.

Improve discussion of the docking site's biological relevance.

Remove extraneous introductory background.

Ensure all compound IDs, docking poses, and data are accurately labeled and interpreted.

Consider additional computational validation (e.g., benchmarking with known ligands or adding entropy calculations).

If the authors can address these concerns and revise accordingly, the study may provide valuable insights for phytochemical-based antiviral drug discovery.

Reviewer #3: Dear Esteemed Editor,

I have examined the publication entitled Exploring Dolichos lablab Compounds as Potential Inhibitors for Fusion (F) Protein of Human Metapneumovirus (HMPV): A Systematic Computational Approach. This manuscript provides a thorough in silico examination of the potential inhibitory effects of phytocompounds derived from Dolichos lablab against the fusion (F) protein of human metapneumovirus (HMPV). The study is pertinent and opportune, as it addresses the unmet clinical need for effective antiviral therapies that specifically target HMPV. In order to identify and assess potential bioactive molecules, the authors have implemented a sequential computational methodology that includes ligand mining, molecular docking, ADMET and bioactivity screening, molecular dynamics (MD) simulation, and MMGBSA binding energy calculations.

However, I have pinpointed multiple areas requiring improvement prior to the manuscript's recommendation for publication

Strengths:

Scope and Relevance:

The study investigates the absence of targeted therapies for HMPV by conducting a screening of phytochemicals derived from Dolichos lablab. This is both novel and pertinent to the global respiratory health. It also incorporates ligand mining, molecular docking, ADMET profiling, bioactivity prediction, molecular dynamics (100 ns), and MMGBSA calculations, offering a comprehensive computational analysis.

The three compounds Brassinolide, Quercetin, and 2-Hydroxygenistein were prominently identified as leading candidates based on consistent outcomes from docking, ADMET, and MD simulations. Comprehensive tables and figures enhance transparency and replicability. The incorporation of MMGBSA values and protein-ligand interaction maps significantly augments scientific rigour.

Methodology:

The author utilized AutoDock Vina, SwissADME, admetSAR, Molinspiration, and Desmond effectively. They also made use of 100 ns molecular dynamics simulations, which are standard or above average in duration for preliminary in silico investigations.

The MMGBSA computation utilizing PRODIGY facilitates energetic validation of the docking results.

Presentation:

The methodologies and software are documented with their respective versions and settings.

Also, the data, tables and figures are appropriately labelled.

Suggestions:

Incorporate RMSD-based validation or re-docking of established inhibitors to ascertain docking precision

Please describe the filtering process.

Can you explain the necessity for subsequent experimental investigations in vitro and in vivo?

Also you should examine Lipinski's rule infractions and their impact on translational viability.

6. PLOS authors have the option to publish the peer review history of their article (what does this mean? ). If published, this will include your full peer review and any attached files.

**Do you want your identity to be public for this peer review?** For information about this choice, including consent withdrawal, please see our Privacy Policy .

Reviewer #1: No

Reviewer #2: No

Reviewer #3: **Yes: ** Ikechukwu Kanu

---

## [Author Response · Author response to Decision Letter 1]

27 Jul 2025

We sincerely thank the editors and reviewers for their thoughtful and constructive feedback. We have carefully addressed all comments and suggestions to improve the clarity, rigor, and scientific quality of the manuscript. Detailed responses to each comment have been provided in the attached point-by-point rebuttal document, and corresponding revisions have been made throughout the manuscript as requested. We hope the revised version meets the expectations and standards of the journal, and we appreciate the opportunity to resubmit our work for further consideration.

Please let us know if any further clarification or modification is needed.

---

## [Decision Letter · Decision Letter 1]

20 Aug 2025

PONE-D-25-32598R1Exploring Dolichos lablab Compounds as Potential Inhibitors for Fusion (F) Protein of Human Metapneumovirus (HMPV): A Systematic Computational ApproachPLOS ONE

Dear Dr. Morshed,

Thank you for submitting your manuscript to PLOS ONE. After careful consideration, we feel that it has merit but does not fully meet PLOS ONE’s publication criteria as it currently stands. Therefore, we invite you to submit a revised version of the manuscript that addresses the points raised during the review process.

**ACADEMIC EDITOR: **Many thanks to the authors for responding positively to the initial concerns. The revision has improved the quality of the submission. However, some grey areas still exist, and these require the authors’ significant attention through another round of revision. In lines 361-382, the authors have subtitled this section including the Table 4 as "Bioactivity". However, the only parameters presented there are TPSA, Molar volume and Rotatable bonds, which are understandably, physicochemical parameters for predicting polarity, size and flexibility of the molecules respectively. The rationale for tagging these "Bioactivity" remains confusing. Moreover, the data are mere predictions which require extensive experimental validation for certainty. I strongly recommend that these section be reconstructed and the superfluous statements be modified. The title should also indicate prediction since no experimental results have been provided. 

We look forward to receiving your revised manuscript.

Kind regards,

Yusuf Oloruntoyin Ayipo, Ph.D

Academic Editor

PLOS ONE

Journal Requirements:

Additional Editor Comments:

Many thanks to the authors for responding positively to the initial concerns. The revision has improved the quality of the submission. However, some grey areas still exist, and these require the authors’ significant attention through another round of revision. In lines 361-382, the authors have subtitled this section including the Table 4 as "Bioactivity". However, the only parameters presented there are TPSA, Molar volume and Rotatable bonds, which are understandably, physicochemical parameters for predicting polarity, size and flexibility of the molecules respectively. The rationale for tagging these "Bioactivity" remains confusing. Moreover, the data are mere predictions which require extensive experimental validation for certainty. I strongly recommend that these section be reconstructed and the superfluous statements be modified. The title should also indicate prediction since no experimental results have been provided.

Reviewers' comments:

Reviewer's Responses to Questions

**Comments to the Author**

1. If the authors have adequately addressed your comments raised in a previous round of review and you feel that this manuscript is now acceptable for publication, you may indicate that here to bypass the “Comments to the Author” section, enter your conflict of interest statement in the “Confidential to Editor” section, and submit your "Accept" recommendation.

Reviewer #1: All comments have been addressed

2. Is the manuscript technically sound, and do the data support the conclusions?

Reviewer #1: Yes

3. Has the statistical analysis been performed appropriately and rigorously? 

Reviewer #1: N/A

4. Have the authors made all data underlying the findings in their manuscript fully available?

Reviewer #1: Yes

5. Is the manuscript presented in an intelligible fashion and written in standard English?

Reviewer #1: Yes

6. Review Comments to the Author

Reviewer #1: (No Response)

7. PLOS authors have the option to publish the peer review history of their article (what does this mean? ). If published, this will include your full peer review and any attached files.

**Do you want your identity to be public for this peer review?** For information about this choice, including consent withdrawal, please see our Privacy Policy .

Reviewer #1: No

---

## [Author Response · Author response to Decision Letter 2]

22 Aug 2025

21 August 2025

Manuscript Number: PONE-D-25-32598R1

Type of manuscript: Research Article

Title: Exploring Dolichos lablab Compounds as Potential Inhibitors for Fusion (F) Protein of Human Metapneumovirus (HMPV): A Systematic Computational Approach.

Dear Editor,

Thank you very much for reviewing our submission titled, “Exploring Dolichos lablab Compounds as Potential Inhibitors for Fusion (F) Protein of Human Metapneumovirus (HMPV): A Systematic Computational Approach”. We thank all reviewers for their very constructive comments and suggestions and revised the manuscript accordingly (marked by track change function in Microsoft word in the revised manuscript). Each reviewer’s comments have been addressed below. We feel these changes have significantly strengthened this manuscript and hope that it will now be suitable for publication in biomedicines.

Editor Comments

Q1] Please review your reference list to ensure that it is complete and correct. If you have cited papers that have been retracted, please include the rationale for doing so in the manuscript text, or remove these references and replace them with relevant current references. Any changes to the reference list should be mentioned in the rebuttal letter that accompanies your revised manuscript. If you need to cite a retracted article, indicate the article’s retracted status in the References list and also include a citation and full reference for the retraction notice.

Response: Thank you for your guidance regarding the reference list. We thoroughly reviewed all references to confirm their accuracy, completeness, and relevance. No retracted articles were identified, and therefore no changes to the reference list were required. The references remain scientifically current and appropriate to the manuscript content.

Additional Editor Comments:

Q1] In lines 361-382, the authors have subtitled this section including the Table 4 as "Bioactivity". However, the only parameters presented there are TPSA, Molar volume and Rotatable bonds, which are understandably, physicochemical parameters for predicting polarity, size and flexibility of the molecules respectively. The rationale for tagging these "Bioactivity" remains confusing.

Response: Thank you for this observation. We agree that the parameters presented in Table 4—TPSA, molar volume, and rotatable bonds—are physicochemical descriptors primarily used to predict molecular polarity, size, and conformational flexibility, rather than direct measures of biological activity. The section was originally titled “Bioactivity” to indicate that these descriptors are often evaluated in early-stage drug-likeness and bioavailability screening, as they influence membrane permeability and receptor binding potential.

However, we acknowledge that the term “Bioactivity” could be misleading in this context. To improve clarity and scientific accuracy, we changed the section title from “Bioactivity” to “Physicochemical Properties Related to Drug-Likeness” in the revised manuscript (page 6, lines 212, 215 and 216, pages 11, 12, lines 362, 364, 366, 387, 390, and 395).

Q2] Moreover, the data are mere predictions which require extensive experimental validation for certainty. I strongly recommend that these sections be reconstructed and the superfluous statements be modified. The title should also indicate prediction since no experimental results have been provided.

Response: Thank you for your insightful suggestions. According to your suggestions, we reconstructed and modified superfluous statements into our revised manuscript (page 12, lines 394, 400-402) as well as we indicated prediction into title section (page 11, line 362).

---

## [Editor Report · Decision Letter 2]

27 Aug 2025

Exploring Dolichos lablab Compounds as Potential Inhibitors for Fusion (F) Protein of Human Metapneumovirus (HMPV): A Systematic Computational Approach

PONE-D-25-32598R2

Dear Dr. Morshed,

We’re pleased to inform you that your manuscript has been judged scientifically suitable for publication and will be formally accepted for publication once it meets all outstanding technical requirements.

Kind regards,

Yusuf Oloruntoyin Ayipo, Ph.D

Academic Editor

PLOS ONE

Additional Editor Comments (optional):

The submission is scientifically sound for publication in this title, and all the concerns raised by the respective reviewers regarding the manuscript quality have been satisfactorily addressed. I hereby recommend the manuscript for publication in the current version.
---

## [Editor Report · Acceptance letter]

PONE-D-25-32598R2

PLOS ONE

Dear Dr. Morshed,

I'm pleased to inform you that your manuscript has been deemed suitable for publication in PLOS ONE. Congratulations! Your manuscript is now being handed over to our production team.

Kind regards,

on behalf of

Dr. Yusuf Oloruntoyin Ayipo

Academic Editor

PLOS ONE